# Comprehensive Systematic Review of Biomarkers in Metastatic Renal Cell Carcinoma: Predictors, Prognostics, and Therapeutic Monitoring

**DOI:** 10.3390/cancers15204934

**Published:** 2023-10-11

**Authors:** Komal A. Dani, Joseph M. Rich, Sean S. Kumar, Harmony Cen, Vinay A. Duddalwar, Anishka D’Souza

**Affiliations:** 1Keck School of Medicine, University of Southern California, Los Angeles, CA 90033, USA; jmrich@usc.edu; 2Eastern Virginia Medical School, Norfolk, VA 23507, USA; sshkumar@ucdavis.edu; 3Children’s Hospital Los Angeles, Los Angeles, CA 90027, USA; 4Norris Comprehensive Cancer Center, University of Southern California, Los Angeles, CA 90033, USA; 5University of Southern California, Los Angeles, CA 90033, USA; hscen@usc.edu; 6Department of Radiology, Keck School of Medicine, University of Southern California, Los Angeles, CA 90033, USA; vinay.duddalwar@med.usc.edu; 7Institute of Urology, University of Southern California, Los Angeles, CA 90033, USA; 8Department of Biomedical Engineering, University of Southern California, Los Angeles, CA 90089, USA; 9Department of Medical Oncology, Norris Comprehensive Cancer Center, University of Southern California, Los Angeles, CA 90033, USA

**Keywords:** renal cell carcinoma, predictive biomarkers, prognostic biomarkers, therapeutic monitoring, immune checkpoint inhibitors, VEGF inhibitors, metastatic disease

## Abstract

**Simple Summary:**

This comprehensive systematic review provides valuable insights into the landscape of biomarkers in clear cell renal cell carcinoma (ccRCC) and their potential applications in the prediction of treatment response, prognosis, and therapeutic monitoring. One of the major challenges in ccRCC is determining the most effective treatment strategies and identifying patients who would benefit from adjuvant or neoadjuvant therapy. This review aims to provide a comprehensive overview of biomarkers in ccRCC and their utility in the prediction of treatment response, prognosis, and therapeutic monitoring in patients receiving systemic therapy for metastatic disease. The findings underscore the importance of incorporating biomarker assessment into clinical practice to guide treatment decisions and improve patient outcomes in ccRCC.

**Abstract:**

Background: Challenges remain in determining the most effective treatment strategies and identifying patients who would benefit from adjuvant or neoadjuvant therapy in renal cell carcinoma. The objective of this review is to provide a comprehensive overview of biomarkers in metastatic renal cell carcinoma (mRCC) and their utility in prediction of treatment response, prognosis, and therapeutic monitoring in patients receiving systemic therapy for metastatic disease. Methods: A systematic literature search was conducted using the PubMed database for relevant studies published between January 2017 and December 2022. The search focused on biomarkers associated with mRCC and their relationship to immune checkpoint inhibitors, targeted therapy, and VEGF inhibitors in the adjuvant, neoadjuvant, and metastatic settings. Results: The review identified various biomarkers with predictive, prognostic, and therapeutic monitoring potential in mRCC. The review also discussed the challenges associated with anti-angiogenic and immune-checkpoint monotherapy trials and highlighted the need for personalized therapy based on molecular signatures. Conclusion: This comprehensive review provides valuable insights into the landscape of biomarkers in mRCC and their potential applications in prediction of treatment response, prognosis, and therapeutic monitoring. The findings underscore the importance of incorporating biomarker assessment into clinical practice to guide treatment decisions and improve patient outcomes in mRCC.

## 1. Introduction

In 2023, there will be an estimated 76,080 new cases and 13,780 new deaths due to kidney cancer in the US [1]. Approximately 90–95% of these neoplasms are renal cell carcinoma (RCC), with 16% presenting with regional spread and another 16% presenting with distant metastasis [2,3]. Clear cell RCC (ccRCC) is the most common subtype, accounting for over 70–80% of RCC [4]. The cure rate is high for patients with early, localized disease, with 5-year survival at more than 90% [5]. In contrast, 5-year survival drops to 12% for patients with distant metastatic disease.

Historically used chemotherapy and radiation therapy lacked sensitivity in ccRCC [6,7], and therefore an improved understanding of the biochemistry and genetic molecular landscape in ccRCC has led to the discovery of novel therapeutic agents [8,9,10]. 

Despite the advances that have been made, there are many challenges to treating ccRCC. In particular, pure anti-angiogenic trials (which have been largely negative) and pure immune-checkpoint monotherapy trials have been applied (with one positive trial so far) to the adjuvant setting with continued uncertainty as to who would benefit from adjuvant therapy or neoadjuvant therapy and for how long [11]. The advent of molecular signatures brings forth the opportunity to better understand how to personalize therapy in ccRCC, enabling clinicians to improve patient outcomes. The purpose of this review is to systematically identify biomarkers that have the potential to diagnose, predict, prognose, and track therapeutic monitoring in ccRCC patients receiving systemic therapy with immune oncology (IO), tyrosine kinase inhibitors (TKI), VEGF inhibitors (VEGFi), or a combination for treatment of adjuvant, neoadjuvant, and metastatic disease. These terms are defined in Table 1 [12].

Prior reviews have broached this topic but have not provided a comprehensive review of the vast variety of biomarkers that exist in ccRCC or have focused on a specific form of therapy. For example, Gulati et al. aim to evaluate validated biomarkers that are being utilized to guide treatment choices and help identify pathways of resistance in other tumor types, and do this by organizing their paper by biomarker (PD-L1, tumor mutational burden, VHL, PBRM1, BAP1, SETD2, a few genomic signatures) [13]. Similarly, Raimondi et al. focus on predictive markers for immunotherapy response [14]. Farber et al. did explore the various serum, urine, imaging, and immunohistological biomarkers that had diagnostic, prognostic, and predictive utility up until 2017. However, this review once again organized their finding by type of biomarker [15]. Our review strove to incorporate all of this content in one place by exploring all biomarkers that have been studied from January 2017 to December 2022 by treatment line.

## 2. Materials and Methods

The protocol of the present systematic review and meta-analysis was following the Preferred Reporting Items for Systematic Reviews and Meta-Analyses (PRISMA) guidelines. The PRISMA checklist was shown in Appendix A. The Covidence platform (reference) was used for paper importing and screening to streamline the review process [16]. This systemic review was conducted on schedule without its registration in PROSPERO.

### 2.1. Data Sources and Searches

Two authors independently searched relevant studies from the PubMed Advanced Search Builder literature database. The retrieval time ended on 31 December 2022. The systematic searching was restricted to the English language and human subjects. The following search strategy was employed in PubMed: ((biomarker) OR (prognostic marker) OR (gene expression)) AND ((immunotherapy) OR (immune checkpoint inhibitor) OR (ICI) OR (targeted therapy) OR (tyrosine kinase inhibitor) OR (TKI) OR (VEGF inhibitor)) AND ((clear cell renal cell carcinoma)) OR (ccRCC) OR (clear cell RCC)) AND ((metastatic) OR (metastasis) OR (metastases) OR (stage 4) OR (stage IV) OR (adjuvant) OR (neoadjuvant)) NOT (“case report”[Title]) NOT (“case reports”[Publication Type]) NOT (review [Publication Type]).

### 2.2. Inclusion and Exclusion Criteria

Two authors independently screened the literature according to the following criteria: (1) studies including patients who were diagnosed with primary or metastatic RCC after cytology or pathology; (2) studies that were Phase I, II, III, or IV or retrospective studies; (3) studies that focused on systemic therapies to treat adjuvant, neoadjuvant, or metastatic disease; (4) studies that used IO, TKI, VEGFi, or a combination of these therapies; and (5) articles that were published in English. For repeatedly published studies, we only chose the latest literature or the literature with the largest sample size. The articles were excluded based on the following reasons: (1) without the normal control group data; (2) review articles, editorials, comments, letters, case reports, etc.; (3) duplicated data; and (4) animal experiment. Only articles published between January 2017 and December 2022 were included, given that RCC treatment protocols have frequently changed. 

### 2.3. Data Extraction and Literature

A standardized data collection form was employed to extract the following information by 2 authors independently: the first author’s name, publication year, title, DOI, abstract, paper methodology, sample size, marker name, marker purpose, marker class, type of maker, drug, drug class, major findings, and limitations. These data can be found in Appendix A. In case no direct HR or 95% CI was provided in the publications or could not be calculated through the existing data, we tried to contact the corresponding author to obtain the relevant data. If no response was received from the author, data were extracted from the survival curve. In case of disagreements during data extraction, a third author would participate in the discussion.

### 2.4. Quality Assessment

The Newcastle–Ottawa quality (NOS) assessment scale was used to evaluate the quality of the included studies by the 2 authors independently. The full mark of the scale was 9. Scores with 0–3, 4–6, and 7–9 were regarded as low quality, moderate quality, and high quality, respectively.

## 3. Results/Discussion

The purpose of this review is to systematically identify biomarkers that have the potential to diagnose, predict, prognose, and track therapeutic monitoring in ccRCC patients receiving systemic therapy with ICI, TKI, VEGFi, or a combination for treatment of adjuvant, neoadjuvant, and metastatic disease. This review organizes our findings first by marker class, then drug class, and finally paper methodology (Figure 1). Various types of biomarkers currently exist including immunologic, genomic, radiogenomic, and physiologic biomarkers. Other markers do not fit into any of these classes. A complete table of all biomarkers and associated data can be found in Appendix A. Additionally, the data collected in this paper can be visualized via Flourish at the following website: https://public.flourish.studio/visualisation/14189718/ (accessed on 22 July 2023). 

### 3.1. Immunologic Biomarkers

Immunologic markers are those that serve as surrogate markers for cellular activation and play an important role in the function of the immune system. These includes serum cytokines, chemokines, adipocytokines, soluble forms of cell receptors, and immune activation markers [17].

#### 3.1.1. Immunotherapy

Immune checkpoint inhibitors are monoclonal antibodies that boost anticancer immune responses by targeting various immune receptors. These agents have been successfully used to treat RCC in clinical trials and/or are being studied in several prospective and retrospective studies. 

#### 3.1.2. Clinical Trials 

There are few clinically proven predictive and prognostic biomarkers for ccRCC treated with first-line therapies, but novel drugs and biomarkers have been emerging from clinical trials. The safety and efficacy of rocapuldencel-T, a type of immunotherapy prepared from mature dendritic cells, was explored in a Phase III study in combination with standard of care sunitinib as a first-line therapy for mRCC. When compared to the standard of care, Rocapuldencel-T produced immune responses in 70% of patients and the strength of the response correlated with OS. High baseline numbers of T regulatory cells are associated with improved outcomes in patients treated with rocapuldencel-T, but were associated with worse outcomes in patients receiving SOC treatment [18]. 

##### Prospective Studies

PD-1 is a receptor found on T-cells which binds to PD-L1, a receptor often upregulated on cancer cells. This binding interaction prevents T-cells from destroying cancer cells, enabling the tumor to escape the immune system. Clinical trials have proven the efficacy of anti-PD1 therapies as a mainstay systemic treatment in ccRCC patients [9,19,20,21]. Therefore, it is natural that PD-1 expression levels can be predictive of treatment response to anti-PD1 therapies. Pignon et al. found that tumor cell PD-L1 expression in combination with PD-1 expression on CD8+ T-cells may predict outcome of nivolumab in mccRCC. Overall PD-L1 expression was not clinically relevant which suggests that in ccRCC, PD-L1 expression on T-cells (but not IC) drives immune evasion and can be reversed by anti-PD-1 therapies [22]. Atkins et al. found that PD-L1 expression has limited clinical utility in selecting patients for nivolumab monotherapy, but may have potential as a predictive biomarker of nivolumab monotherapy efficacy within a multifactorial predictive biomarker model [23]. 

By contrast, Mahoney et al. examined soluble PD-L1 (sPD-L1) as a biomarker within serum, rather than from tumor tissue. They found that sPD-L1 is a marker of nivolumab-refractory disease in RCC and is more complex than just a substitute for PD-L1. This is because aggressive disease produces sPD-L1, but there may be a distinct secondary pathway by which some patients with some or complete response to nivolumab produce sPD-L1. Therefore, a comparison of baseline and on-therapy sPD-L1 levels in RCC may be able to predict progressive disease in patients taking nivolumab [24]. Incorvaria et al. showed that the plasma levels of sPD-1, sPD-L1, and sBTN3A1 can predict improved response to second-line nivolumab. Incorvaria et al. analyzed the dynamic changes of plasma after nivolumab treatment and found a statistically significant decrease of sPD-1 after 28 days of therapy in the long-responder patients [25]. 

Ross-Macdonald et al. found that non-response to nivolumab was correlated with tumors highly infiltrated with T-cells. The degree of infiltration was measured at baseline and on day 28 by a T-cell receptor “CD3TCR” expression score using a gene set of the CD3-γ, CD3-δ, CD3-ε, CD3-ζ, TCR-α, and TCR-β subunits [26]. They also found that IL-18 mRNA was differentially upregulated in treatment responders [26]. Similarly, Chehrazi-Raffle found that patients with clinical benefit from immunotherapy had higher levels of interferon-γ and IL-12 [27]. 

Other predictive markers being explored include circulating endothelial cells. García-Donas et al. found that, when given antiangiogenic treatments (sunitinib and/or pazopanib), the detection of higher CEC levels, defined as DAPI+, CD105+, and CD45−, was associated with progression-free survival in ccRCC. The study also found that CEC levels did not change significantly despite tumor progression compared with baseline, which suggests that CECs could be determined at any time during treatment, remaining a stable predictive biomarker [28]. 

Prognostic liquid biopsy biomarkers have been explored prospectively as well. Bootsma et al. examined circulating tumor cell abundance and HLA I to PD-L1 (HP) ratio as prognostic markers in RCC and demonstrated that both markers can be used to monitor treatment response Specifically, the direction in which CTC enumeration changes was strongly associated with OS and increases in the HP ratio over time collated with worse outcomes [29]. Billon et al. suggested that the baseline level of plasmatic BTN2A1 could be an independent prognosis factor of PFS for second-line nivolumab after a TKI in patients with mRCC [30]. Incorvaria et al. also studied this biomarker, BTN2A1 was not significantly associated with OS or PFS [25]. However, Billon et al. found that patients with PFS > 18 months seemed to have lower levels of sBTN2A1 than patients with PFS < 18 months [30]. In a study that analyzed 106 immune cell populations in fresh blood, Carril et al. found that baseline unswitched memory B cells (NSwM B cells) were increased in responders and associated with improved OS and PFS in patients on second-line nivolumab following prior nivolumab therapy. They also found that BCA-1/CXCL13 and BAFF, which are chemokines strongly expressed in the secondary lymphoid organs, were both associated with worse OS and inversely correlated to NSwM B cells [31]. De Giorgi et al. studied the association between inflammation in general (and Body Mass Index, or BMI) with the clinical outcome and found that a normal BMI combined with inflammation tripled the risk of death, suggesting that these biomarkers are critical prognostic factors for OS in patients with RCC treated with nivolumab. In univariate analysis, markers such as systemic immune-inflammation index (SII), neutrophil to lymphocyte ratio (NLR), and platelet to lymphocyte ratio (PLR) were able to predict outcome; and in multivariate analyses, SII ≥ 1375, BMI < 25 kg/m^2^, and age ≥ 70 years independently predicted overall survival. SII changes at 3 months also predicted OS [32]. 

A study by Saal et al. aimed to evaluate the prognostic value of the modified Glasgow prognostic score (mGPS) in patients with mRCC treated with ICIs, specifically in the context of the IMmotion151 trial, which compared atezolizumab plus bevacizumab to sunitinib. The mGPS assigns points based on elevated serum C-reactive protein (CRP) and decreased serum albumin levels. Patients are categorized as low, intermediate, or high risk. The results demonstrated that the mGPS outperformed the International Metastatic Renal Cell Carcinoma Database Consortium (IMDC) score, the current standard for risk stratification in mRCC. The mGPS had a higher concordance index and identified a larger proportion of patients as low risk while maintaining comparable survival rates. This study suggests that the mGPS could replace the IMDC score as a simple and effective prognostic tool in the era of immuno-oncology for mRCC patients [33].

##### Retrospective Studies

Retrospective studies disclose even more unique biomarkers that have both predictive and prognostic value when studying effects of immunologic treatments. Many biomarkers have been studied, the most common of which are C-reactive protein (CRP), NLR, and immune-related adverse events (irAEs). Many other markers were also studied and can be found in Appendix A.

One of the most commonly studied predictive and prognostic markers is CRP, which is a nonspecific marker of inflammation found in blood. Kankkunen et al., Roussel et al., and Ito et al. all studied elevated baseline CRP as a marker of poor prognosis in RCC, and found that this marker can predict worse OS and PFS on nivolumab [34,35,36]. Kankkunen also found on multivariate analyses that patients with elevated baseline and on-treatment CRP had shorter OS and PFS than patients with normal CRP [34]. Similarly, Ito et al. found that pretreatment C-reactive protein >3.0 mg/dl was an independent predictor for PFS [36].

Abuhelwa et al. also studied CRP as a prognostic maker in ccRCC. They found that CRP could be used to stratify patients into four levels, with CRP levels inversely correlated with OS and PFS [37]. Fukuda et al. studied the CRP flare response by categorizing patients as CRP flare responders, CRP responders, and non-CRP responders depending on if CRP levels more than double compared with baseline within 1 month after initiation of nivolumab (flare) and then decreased below baseline within 3 months, if CRP levels decreased by ≥30% within 3 months without “flare”, or if CRP levels were minimally changed or unchanged, respectively. They found that the CRP-flare response was associated with significant tumor shrinkage and improved survival outcomes [38]. Takamatsu et al. found that serum CRP higher than 0.5 mg/dl in RCC patients after first-line treatment termination could be a marker of poor prognosis in intermediate-risk patients treated with second-line treatment, given that the median OS of elevated and non-elevated CRP group was 11.5 and 29.4, respectively [39].

Others studied CRP as a predictor of treatment response in RCC. Noguchi et al. found that CRP level at 1 month after treatment with nivolumab may be clinically useful in quickly predicting treatment effect [40]. Fukuda et al. and Klümper et al. both studied the predictive value of the CRP flare response described above [38,41]. Fukuda et al. suggested that the CRP flare response in nivolumab treatment may be related to changes caused by the cancer-immune system within the tumor microenvironment [38]. Klümper et al. expanded on this idea and found that CRP responders, especially CRP flare responders, had significantly prolonged progression-free survival (PFS) compared with non-CRP responders and long-term response (≥12 months) to first-line IO combination therapy [41]. Beyond CRP, several cytokines are being studied as predictive and prognostic biomarkers for immunotherapies, including PD-1, PD-L1, IL6, IL-8, and IL-12. Details regarding these studies can be found in Appendix A. 

Another commonly studied predictive and prognostic biomarker is NLR, which is a biomarker that links the two parts of the immune system: the innate immune response, mediated by neutrophils, and adaptive immunity, supported by lymphocytes [42]. There is some conflicting evidence on the clinical significance of this marker. Roussel et al., Lalani et al., and Jeyakumar et al. found that higher baseline NLR predicted worse OS and PFS on nivolumab in mccRCC patients [35,43,44], while Nishiyama et al. found that baseline NLR was not associated with OS or PFS. Specifically, Jeyakumar et al. concluded that NLR ≥3 prior to initiating ICI therapy was an independent predictor of OS and PFS [44], while Ito et al. and Nishiyama et al. instead found that NLR of ≥3 at 4 weeks of nivolumab therapy was an independent predictor of OS and PFS [36,45]. Lalani’s and Ikarashi’s work supports Nishiyama’s findings as they both also found that higher NLR at 6 weeks was a significantly stronger predictor of all three outcomes than baseline NLR [43,46]. Zahoor et al. studied NLR as a predictor of progressive disease (i.e., metastasis), and found that the risk of progressive disease was elevated with higher baseline NLR [47]. Beyond NLR, there are several immune cell populations being studied as predictive and prognostic biomarkers for immunotherapies, including lymphocyte counts, neutrophil counts, PLR, neutrophil to eosinophil ratio (NER), monocyte to eosinophil ratio (NER), monocyte to lymphocyte ratio (MLR). Details regarding these studies can be found in Appendix A. 

irAEs, a known side effect associated with ICIs, are a set of autoimmune conditions that can affect any organ in the body, which makes them difficult to diagnose and manage [48]. Clinicians have theorized that the presence of irAEs may have some implications for mRCC prognosis but the relationship is fairly unclear. Kankkunen et al. and Martini et al. found that irAEs, particularly thyroid irAEs, had significantly improved clinical outcomes in mRCC patients treated with ICIs as second-line therapy [34,49]. Ikeda et al. and Ishihara et al. both studied the relationship between irAEs and PFS and OS in patients treated with nivolumab. Ikeda et al. found that irAE development was significantly associated with PFS but not with OS [50], while Ishihara et al. found that irAE development was significantly associated with both PFS and OS [51]. These findings suggest that irAEs may be used as a clinical biomarker predicting treatment outcome in mRCC patients treated with ICIs. 

#### 3.1.3. VEGFR TKIs

TKIs are a class of drugs that inhibit signal transduction of protein kinases which in turn critically disrupt cellular signaling [52]. One group of growth factor receptors affected by TKIs are VEGF receptors (VEGFRs), which have been known to play a key role in the angiogenesis caused by cancers, including RCC [53]. 

VEGFR TKIs form a class of adjuvant therapy used in RCC. Drugs that are approved by the US FDA include sorafenib, sunitinib, axitinib, and pazopanib [53]. These agents have been successfully used to treat RCC in clinical trials and biomarkers related to VEGFR TKI efficacy and prognosis are being studied in several prospective and retrospective studies. 

##### Prospective Studies

Prospective studies have studied both the predictive and prognostic abilities of angiogenesis markers as well as plasma cytokines in patients treated with VEGFR TKIs. Mauge et al., Jilaveanu et al., Oudard et al., and Xu et al. all studied the prognostic abilities of various angiogenesis markers [54,55,56,57] in patients receiving first-line sunitinib or pazopanib. In this multicenter, prospective, open-label, phase II trial, Mauge et al. included treatment-naïve patients with mccRCC who had received two cycles of sunitinib before nephrectomy, and found that baseline values of angiogenesis markers (found in parentheses) were significantly associated with a change in primary renal tumor size (endothelial progenitor cells), PFS (vascular endothelial growth factor-A, stromal cell-derived factor(SDF)-1, soluble VEGF receptors (sVEGFR) 1 and 2), and OS (SDF-1 and sVEGFR1). They also found that changes in the following markers (found in parentheses) during treatment were significantly associated with a change in primary renal tumor size (SDF-1 and platelet-derived growth factor-BB), PFS (sVEGFR2), and OS (SDF-1 and sVEGFR1) [54]. Jilaveanu et al. focused on the prognostic of changes in microvessel density (MVD) in patients taking adjuvant sunitinib and sorafenib. They found that, on both univariate and multivariate analyses, high MVD (defined as above the median) was associated with increased OS as compared to patients receiving placebo, and there was a less significant association when comparing populations of patients treated with sunitinib or sorafenib [55]. Oudard et al. also looked at VEGFR-1 when they studied the prognostic value of subpopulations of pro-angiogenic monocytes and found that a more than 20% reduction from baseline value of VEGFR-1+CD14 monocytes at 20 weeks after starting sunitinib or pazopanib was associated with a significant increase in PFS and OS. They also found that more than 20% reduction from baseline value of Tie2 + CD14 cells monocytes was associated with a significant increase in OS [56]. Similarly, Xu et al. found that sVEGFR-2 decreased after both 4 and 6 weeks of treatment on sunitinib or sorafenib and that sFLT-1 decreased after 4 weeks on sunitinib and 6 weeks on sorafenib [57]. 

Jilaveanu et al. also studied the predictive value of MVD. They also found that high MVD significantly correlated with Fuhrman grade 1–2, clear cell histology, and absence of necrosis but not with gender, age, sarcomatoid features, lymphovascular invasion, or tumor size, and therefore concluded that MVD was a better potential as a prognostic marker, rather than a predictive marker [55]. Hakimi et al. also studied the predictive value of angiogenesis markers, as well as mutation profiles and macrophage infiltration, in patients receiving received first-line sunitinib or pazopanib and found that patients with higher angiogenesis scores had a superior outcome independent of the IMDC risk category. They also attributed the predictive capabilities of angiogenesis markers to upregulation and suppression of angiogenesis observed with loss-of-function mutations in PBRM1 and BAP1, respectively [58]. 

Other groups have prospectively looked at the predictive and prognostic abilities of various plasma cytokines in patients treated with VEGFR TKIs. IL-6 is a plasma cytokine that is known to play a pathologic role in chronic inflammation as well as RCC [59,60]. Pilskog et al. studied the ability of IL-6 to prognose PFS and OS and predict response to sunitinib [61,62] while Chehrazi-Raffle et al. focused on IL-6′s predictive abilities [27]. In Pilskog et al.’s first study, they evaluate both the prognostic and predictive ability of interleukin-6 receptor α (pIL6Rα) in mRCC patients treated with sunitinib, and found that low tumor expression of IL6Rα, which is directly correlated with expression of some angiogenesis markers, was significantly associated with improved response to sunitinib [61]. Pilskog et al.’s second study focused on the predictive ability of plasma interleukin-6 (pIL6), pIL6Rα, and interleukin-6 signal transducer (pIL6ST) in mRCC patients treated with sunitinib. In this study, they found that low pIL6 at baseline was also significantly associated with improved PFS and response to sunitinib. Furthermore, patients with high pIL6ST at baseline showed significantly improved OS. This signified that pIL6 may have both prognostic and predictive potential in mRCC. Chehrazi-Raffle et al. confirmed this finding when they found that patients treated with VEGF-TKIs had lower pretreatment levels of interleukin-6 (IL-6) (as well as IL-1RA and granulocyte CSF). This study, however, did support IL-6′s ability to predict treatment response and instead found that clinical benefit from VEGF TKIs could be monitored with decreases in IL-13 and granulocyte macrophage CSF as well as increases in VEGF at one month [27].

Like Chehrazi-Raffle et al., Bellmunt et al. and Xu et al. all looked at the predictive and prognostic value of plasma cytokines other than IL-6. Bellmunt et al. studied the predictive value of IL-10 levels in mRCC patients receiving second-line pazopanib after failure of a prior TKI. They found that lower circulating levels of IL-10 were observed in responding patients at 8 weeks after treatment [63]. Xu et al. investigated the effects of adjuvant VEGFR TKIs on circulating cytokines and found that when on sunitinib and sorafenib, CXCL10 elevated at 4 and 6 weeks was associated with worse DFS [57]. 

Zizzari et al. and Montemagno et al. both looked at soluble forms of PD-L1 and PD-1 as prognostic and predictive markers of sunitinib efficacy. Zizzari et al. identified soluble PD-L1 and PD-1 as two of seven soluble immune molecules (IFNγ, sPDL2, sHVEM, sPD1, sGITR, sPDL1, and sCTLA4) as well as CD3 + CD8 + CD137+ and CD3 + CD137 + PD1 + T-cell populations as markers modulated by TKI therapy [64]. Montemagno specifically studied first-line sunitinib and bevacizumab, and confirmed that levels of soluble PD-L1 and PD-1 correlated with PFS with sunitinib only. They found that sunitinib treated patients with high baseline plasmatic levels of sPD-L1 and sPD-1 had a shorter PFSm [65]. There are several other retrospective studies that have been conducted on PD-L1 and PD-1, and can be found in Appendix A. 

##### Retrospective Studies

Retrospective studies identified even more unique potential biomarkers that have both predictive and prognostic value when studying the effects of immunologic treatments. Many biomarkers have been studied, the most common of which are CRP and NLR. Many other markers were also studied and can be found in Appendix A.

Several retrospective studies have investigated the prognostic value of CRP levels in mRCC patients and their response to TKIs. Takamatsu et al. investigated the prognostic value of baseline CRP level in intermediate-risk mRCC patients treated with first-line VEGFi therapy. They reported that higher baseline CRP levels were associated with inferior OS and PFS outcomes [66]. They then performed a second study in which they found that elevated baseline serum CRP levels prior to second-line treatment in intermediate-risk mRCC patients was also associated with poor prognosis [39]. Both studies showed that intermediate-risk mRCC patients could be divided into two prognostic subgroups [39,66]. Similarly, Wang et al. identified dynamic changes in the systemic inflammatory response, including CRP, that could be prognostic indicators in mRCC. They observed that patients with elevated CRP levels at baseline and during treatment had significantly worse OS and PFS rates compared to those with normal CRP levels [67]. Takagi et al. also confirmed this finding with their study aiming to identify prognostic markers, including CRP, for refined stratification of intermediate-risk ccRCC patients treated with first-line TKI therapy. They also found that elevated CRP levels were associated with worse OS and PFS outcomes in this patient population [68]. Furthermore, Teishima et al. observed that normalization of CRP levels following cytoreductive nephrectomy in mRCC patients treated with TKIs was associated with improved overall survival. They found that patients who achieved CRP normalization after surgery had better OS rates compared to those who did not [69].

Other studies aimed to show CRP’s efficacy as a predictive marker. Yasuda et al. demonstrated that early response of CRP levels could predict survival in patients with mRCC undergoing TKI treatment. They found that patients who achieved a significant decrease in CRP levels within one month of TKI therapy initiation had improved OS and PFS [70]. Klümper et al. explored the CRP flare response as a predictor of long-term efficacy in first-line anti-PD-1-based combination therapy for mRCC. They observed that patients who exhibited a CRP flare response had significantly prolonged PFS and higher rates of long-term response [41]. In addition to these studies, Erdogan et al. investigated the association between early changes in systemic inflammatory markers, including CRP, and treatment response in patients receiving pazopanib. They observed that patients who showed a significant decrease in CRP levels within one month of pazopanib treatment initiation had improved clinical outcomes [71]. Collectively, these studies highlight the importance of CRP as a prognostic and predictive marker and highlight the role it may play in mRCC patients receiving TKIs in the future. Beyond CRP, there are several cytokines being studied as predictive and prognostic biomarkers for immunotherapies, including PD-1, PD-L1, HIF-1α, IL-8, and FGFR2. Details regarding these studies can be found in Appendix A. 

#### 3.1.4. Combination Therapy

Combination therapy involves the use of multiple drug classes to enhance the efficacy of anti-cancer treatments. This approach potentially reduces drug resistance, while reducing tumor growth, mitotically active and cancer stem cells, and metastatic potential [72]. In RCC, the current gold standard is antiangiogenic agents combined with tyrosine kinase, mTOR, or immune checkpoint inhibitors [73]. These agents have been successfully used to treat RCC in clinical trials and biomarkers related to combination therapy efficacy and prognosis are being studied in several prospective and retrospective studies. 

##### Prospective Studies

Several studies have provided valuable insights into the clinical activity and molecular correlates of response to immunotherapy in renal cell carcinoma (RCC) patients. McDermott et al. found that atezolizumab–bevacizumab combination therapy showed improved clinical activity and higher overall response rates compared to sunitinib, providing molecular characteristics of each treatment option. The study also identified specific molecular markers associated with response to immunotherapy, such as high tumor mutational burden and PD-L1 expression. These findings provide important information for patient selection and treatment decision making [74].

In a study by Martini et al. the focus was on angiogenic and immune-related biomarkers in patients receiving axitinib/pembrolizumab treatment. The researchers observed that specific angiogenic and immune-related biomarkers, including angiopoietin-2 and vascular endothelial growth factor, were associated with treatment response and outcomes. This highlights the potential of these biomarkers as predictive factors for therapy response and the importance of considering both angiogenesis and immune pathways in RCC treatment strategies [75].

Msaouel et al. conducted a phase 1–2 trial evaluating sitravatinib and nivolumab in clear cell RCC patients who had progressed on antiangiogenic therapy. The results demonstrated promising clinical activity, with a significant proportion of patients experiencing tumor shrinkage and disease control. The study also highlighted the potential of sitravatinib to overcome resistance to antiangiogenic therapy and enhance the efficacy of immunotherapy. These findings suggest that the combination of sitravatinib and nivolumab could be a promising treatment approach for refractory RCC patients [76].

##### Retrospective Studies

In a study by Kamai et al. the researchers investigated the expression of adenosine 2A receptors (A2AR) in metastatic renal cell carcinoma (RCC) and its impact on treatment response and patient survival. The study revealed that increased expression of A2AR was associated with poorer response to anti-VEGF agents and anti-PD-1/anti-CTLA4 antibodies. Patients with higher A2AR expression levels demonstrated shorter overall survival rates, indicating the potential of A2AR as a prognostic biomarker in metastatic RCC. These findings highlight the significance of A2AR in the immunotherapy response and suggest that targeting A2AR signaling may improve treatment outcomes in RCC patients [77]. The study contributes to the understanding of the molecular mechanisms underlying immunotherapy resistance and provides insights for the development of novel therapeutic strategies in metastatic RCC. Beyond A2AR, there are several markers being studied for their predictive and prognostic effects in patients taking immunotherapies. Details regarding these studies can be found in Appendix A. 

### 3.2. Genomic Biomarkers

Genomic markers are DNA or RNA sequences that are known to cause disease or increase susceptibility to disease. This marker class is the basis of genetic testing, which identifies changes in chromosomes, genes, and proteins [78], as well as companion diagnostics that match patients to a specific drug or therapy based on identified genetic changes or genomic markers. 

#### 3.2.1. Immunotherapy

##### Prospective Studies

Several studies have investigated predictive biomarkers for immunotherapy response in metastatic renal cell carcinoma (mRCC) patients. Incorporvaia et al. and Epaillard et al. specifically studied PD-1 and PD-L1 as genomic biomarkers that may predict response to nivolumab. Incorporvaia et al. identified a “Lymphocyte MicroRNA Signature” as a potential predictive biomarker of immunotherapy response and plasma PD-1/PD-L1 expression levels in mRCC patients, suggesting a role for epigenetic reprogramming in treatment outcomes. They found that patients with higher levels of specific microRNAs had better response rates and improved overall survival [79]. Epaillard et al. also investigated the efficacy of nivolumab and ipilimumab in treatment-naïve patients with metastatic kidney cancer. This phase 2 biomarker-driven trial found that patients with PD-L1 positive tumors exhibited higher response rates and improved progression-free survival when treated with nivolumab and ipilimumab combination therapy [80].

Miao et al. and Ross-Macdonald et al. studied other genomic biomarkers that may predict response to nivolumab. Miao et al. explored the genomic correlates of response to immune checkpoint therapies in clear cell RCC and identified specific genomic features associated with treatment response, including mutations in the PBRM1 gene. They observed that patients with PBRM1 mutations had a higher treatment response rate and longer progression-free survival [81]. Ross-Macdonald et al. investigated molecular correlates of response to nivolumab at baseline and during treatment and identified several gene expression signatures associated with clinical response. They suggested that these gene signatures could serve as potential predictive biomarkers for nivolumab therapy [26]. Kim et al. explored the potential of circulating tumor DNA as a predictor of therapeutic responses to immune checkpoint blockades in mRCC. They observed that the detection of specific mutations in circulating tumor DNA was associated with better treatment responses and improved progression-free survival [82].

##### Retrospective Studies

Several genomic markers are being studied as predictive and prognostic potential in immunotherapies. Details regarding these studies can be found in Appendix A. 

#### 3.2.2. VEGF TKIs

##### Prospective Studies

Several studies have investigated the role of biomarkers in predicting treatment response and prognosis in advanced renal cell carcinoma (RCC) patients receiving sunitinib therapy. In a study by Dietz et al. molecular alterations specific to individual patients with metastatic ccRCC were identified, and these alterations were found to be associated with disease progression despite TKI therapy. The results underscore the need for personalized treatment approaches that consider the unique molecular profiles of patients to improve therapeutic outcomes [83]. Examining the hypoxia-inducible factor (HIF) pathway and c-Myc as potential biomarkers, Maroto et al. assessed their predictive value in response to sunitinib treatment in metastatic ccRCC. Their findings indicated that the status of these biomarkers could help identify patients who are more likely to respond favorably to sunitinib therapy, enabling a more targeted approach to treatment selection [84]. Wierzbicki et al. evaluated the prognostic significance of several biomarkers, including VHL, HIF1A, HIF2A, VEGFA, and p53, in ccRCC patients treated with sunitinib as a first-line therapy. The study revealed that the expression levels of these biomarkers were associated with clinical outcomes, providing valuable prognostic information [85]. Nayak et al. investigated the role of circulating tumor cells (CTCs) as biomarkers in metastatic ccRCC. They found that the presence of CTCs was associated with disease status and provided valuable information regarding treatment response. The study suggested that CTCs could serve as non-invasive markers for monitoring disease progression and guiding treatment decisions in ccRCC [86].

Other studies have looked at the role of biomarkers in predicting treatment response and prognosis in patients receiving sorafenib therapy. Gudkov et al. conducted a study aiming to develop a gene expression-based signature capable of predicting the response to sorafenib in kidney cancer patients. The findings demonstrated that the signature successfully predicted the efficacy of sorafenib treatment, providing valuable insights for personalized therapeutic decision making in renal cell carcinoma (RCC) patients [87]. Crona et al. investigated genetic variants of VEGFA and FLT4 as determinants of survival in RCC patients treated with sorafenib. The study revealed that specific genetic variations in these genes were associated with patient survival outcomes, highlighting the potential of genetic profiling as a prognostic tool to identify individuals who may benefit the most from sorafenib therapy [88].

Bevacizumab was also a commonly explored drug; in a study by Dorff et al. the efficacy of bevacizumab, either alone or in combination with TRC105, was evaluated in patients with refractory metastatic RCC. The results demonstrated the potential clinical benefit of bevacizumab-based therapy in this patient population, suggesting its relevance as a treatment option for refractory metastatic RCC [89]. Bamias et al. conducted a clinical and biomarker study assessing the combination of bevacizumab and temsirolimus as a second-line therapy in advanced RCC patients who had received prior anti-VEGF treatment. The study revealed promising results, with improved clinical outcomes and biomarker profiles observed in patients treated with combination therapy, indicating its potential as an effective therapeutic approach for advanced RCC [90].

##### Retrospective Studies 

Several genomic markers were studied as predictive and prognostic potential in VEGF-TKI therapies. Details regarding these studies can be found in Appendix A. 

#### 3.2.3. mTORi

mTOR inhibitors (mTORi) are chemotherapy drugs that deactivate the PI3K/AKT/mTOR signaling pathway, which in turn prevents tumor angiogenesis and downregulates the expression of hypoxia-inducible factors in RCC. Currently, two mTOR inhibitors are approved for use in metastatic RCC: temsirolimus and everolimus. These agents have been successfully used to treat RCC in clinical trials and biomarkers related to mTORi efficacy and prognosis are being studied in several prospective and retrospective studies, and are summarized in Appendix A. 

##### Prospective Studies

Several studies have focused on identifying prognostic and predictive biomarkers for mTOR inhibitors. Palomero et al. conducted a study investigating the role of EVI1, a transcription factor involved in cell proliferation and differentiation, as a prognostic and predictive biomarker in ccRCC. They found that higher EVI1 expression was associated with worse prognosis and resistance to mTORi therapy, indicating its potential as a valuable biomarker for guiding treatment decisions in ccRCC patients [91].

Zeuschner et al. aimed to identify predictive biomarkers for everolimus treatment in second-line metastatic ccRCC. Through their research, they identified thrombospondin-2 and lactate dehydrogenase (LDH) as potential predictive biomarkers associated with treatment response to everolimus. This suggests that assessing the levels of these biomarkers could aid in patient selection for everolimus therapy, allowing for more personalized and effective treatment strategies.

In addition, Voss et al. investigated the correlation between PTEN expression and treatment outcome in RCC patients receiving everolimus. They found that PTEN expression, rather than mutation status in TSC1, TSC2, or mTOR, was significantly associated with treatment response to everolimus. This highlights the importance of assessing PTEN expression levels as a potential predictive biomarker for everolimus therapy in RCC [92].

##### Retrospective Studies

Flaifel et al. conducted an analysis of the randomized clinical trials METEOR and CABOSUN to investigate the relationship between programmed death-ligand 1 (PD-L1) expression and clinical outcomes in mRCC patients treated with cabozantinib, everolimus, and sunitinib. The study aimed to determine whether PD-L1 expression levels could serve as predictive biomarkers for treatment response. The analysis revealed that PD-L1 expression was not significantly associated with clinical outcomes in terms of progression-free survival and overall survival in patients receiving any of the three treatments. These findings suggest that PD-L1 expression may not be a reliable biomarker for predicting response to cabozantinib, everolimus, or sunitinib in mRCC patients [93].

Several other genomic markers were studied as predictive and prognostic potential in mTORi therapies. Details regarding these studies can be found in Appendix A. 

#### 3.2.4. Combined Biomarkers

##### Prospective Studies 

Both Motzer et al. and McDermott et al. studied gene expression signatures as a biomarker in mRCC by comparing atezolizumab alone or in combination with bevacizumab versus sunitinib. Motzer et al.’s molecular analysis revealed that immune-related gene expression signatures were associated with better outcomes for the combination therapy, while angiogenesis-related signatures showed better outcomes with sunitinib [94]. McDermott et al.’s results revealed that the combination therapy of atezolizumab plus bevacizumab demonstrated improved clinical outcomes, including overall survival, compared to sunitinib alone. Molecular analysis further identified immune-related gene expression signatures associated with better responses to the combination therapy, emphasizing the importance of molecular profiling for guiding treatment decisions in RCC patients.

##### Retrospective Studies

Several genomic markers were studied as predictive and prognostic potential in combination therapies. Details regarding these studies can be found in Appendix A. 

### 3.3. Radiologic Biomarkers

Radiomics is an emerging field of study in which advanced imaging strategies provide structural and phenotypic biomarkers related to key disease processes. Radiomics-based biomarkers are being used to deeply analyze pathophysiologic processes and have provided insights to better diagnose, classify, stratify, and prognosticate tumors, and to assess their response to therapy [95]. 

#### 3.3.1. ICI

##### Prospective Studies

In recent years, there has been growing interest in identifying radiographic biomarkers that can predict clinical outcomes in mRCC patients receiving immune checkpoint inhibitors (ICIs). Tabei et al. conducted a study to assess the early predictive value of 18F-2-fluoro-2-deoxyglucose positron emission tomography/computed tomography (18F-FDG PET/CT) in patients with ccRCC who received nivolumab, an immune checkpoint inhibitor. The study aimed to determine whether metabolic response assessed by 18F-FDG PET/CT after two cycles of nivolumab treatment could predict short-term outcomes. The findings revealed that patients treated with nivolumab had significantly longer progression-free survival and overall survival if they showed a metabolic response on 18F-FDG PET/CT. This suggests that early assessment with 18F-FDG PET/CT could serve as a valuable tool for predicting treatment response and patient outcomes in ccRCC patients receiving nivolumab therapy [96].

Drljevic-Nielsen et al. investigated the potential of spectral dual-layer detector CT parameters as imaging biomarkers in patients with metastatic renal cell carcinoma (mRCC) who were treated with TKIs and immunotherapies. Their first study aimed to assess whether early reduction in spectral dual-layer CT parameters could serve as favorable prognostic indicators. They found that early reduction iodine concentration and effective atomic number predicted significantly longer OS and PFS. Their second study continued the focus on CT parameters, corroborating the predictive value of iodine concentration and adding CT parameters, such as virtual monochromatic images at specific energy levels, as a predictive biomarker. These findings suggest that both pretreatment and spectral dual-layer CT parameters have prognostic utility and could potentially aid in risk stratification and treatment decision making in mRCC patients treated with different therapies [97,98].

##### Retrospective Studies 

Martini et al. conducted a study to evaluate the potential of body composition variables as radiographic biomarkers in this patient population. They found that lower skeletal muscle index and higher visceral adipose tissue area were associated with worse clinical outcomes, including shorter overall survival and progression-free survival. They also found that lower skeletal muscle index was associated with higher levels of systemic inflammation, as indicated by elevated CRP levels. In addition, higher visceral adipose tissue area was linked to greater tumor burden and increased tumor inflammation. These findings suggest that body composition may influence treatment response and prognosis through its impact on systemic inflammation and tumor characteristics [99].

Malone et al. conducted a study to develop a predictive radiomics signature for treatment response to nivolumab in patients with advanced renal cell carcinoma (RCC). Using radiomics analysis, they extracted quantitative imaging features from pre-treatment computed tomography (CT) scans and developed a radiomics signature. The results showed that the radiomics signature was significantly associated with treatment response to nivolumab. This finding suggests that the radiomics signature has the potential to serve as a non-invasive predictive biomarker for treatment response in patients with advanced RCC undergoing nivolumab therapy [100].

Mittlmeier et al. investigated the utility of 18F-PSMA-1007 PET/CT for response assessment in patients with metastatic renal cell carcinoma undergoing TKI or checkpoint inhibitor therapy. The study evaluated the preliminary results of using PSMA-based positron emission tomography/computed tomography (PET/CT) imaging to assess treatment response. The findings indicated that 18F-PSMA-1007 PET/CT showed promising potential in monitoring response to therapy in patients with metastatic RCC. This suggests that PSMA-based PET/CT imaging may serve as a valuable tool for response assessment in RCC patients receiving TKI or checkpoint inhibitor therapy [101].

Zheng et al. compared radiological tumor response assessment based on immune Response Evaluation Criteria in Solid Tumors (iRECIST) and Response Evaluation Criteria in Solid Tumors (RECIST) 1.1 in metastatic clear-cell renal cell carcinoma (ccRCC) patients treated with programmed cell death-1 (PD-1) inhibitor therapy. The study aimed to determine the agreement between the two response evaluation criteria and assess their correlation with overall survival. The results showed that iRECIST provided a more accurate response assessment and better prediction of overall survival compared to RECIST 1.1 in metastatic ccRCC patients treated with PD-1 inhibitor therapy. These findings highlight the importance of using iRECIST as a more reliable and clinically relevant tool for response evaluation in this patient population [102].

Park et al. conducted a study to evaluate the use of computed tomography (CT) texture analysis as a predictive tool for clinical outcomes in patients with mRCC treated with immune checkpoint inhibitors (ICIs). They analyzed CT images of mRCC patients and extracted various texture features to assess tumor heterogeneity. The study found that certain CT texture features were significantly associated with treatment response, progression-free survival, and overall survival in patients receiving ICIs. These findings suggest that CT texture analysis could serve as a non-invasive and promising tool for predicting clinical outcomes and guiding treatment decisions in mRCC patients undergoing ICI therapy [103].

#### 3.3.2. VEGF TKIs

##### Prospective Studies

Udayakumar et al. conducted a study aimed at deciphering the intratumoral molecular heterogeneity in ccRCC using a radiogenomics platform. They integrated radiomic and genomic data to gain insights into the diverse molecular subtypes within ccRCC tumors. The study demonstrated that the radiogenomics approach could identify distinct molecular subtypes based on imaging features, providing valuable information for personalized treatment strategies. The findings highlight the potential of utilizing radiogenomics to improve the understanding of intratumoral heterogeneity in ccRCC and guide precision medicine approaches [104].

Nakaigawa et al. investigated the predictive value of fluorodeoxyglucose positron emission tomography/computed tomography (FDG PET/CT) after the first molecular targeted therapy in patients with renal cell carcinoma (RCC). The study aimed to determine whether FDG PET/CT could serve as a prognostic tool to predict patient survival. The results demonstrated that FDG PET/CT findings after the initial targeted therapy were significantly associated with overall survival in RCC patients. This suggests that FDG PET/CT imaging can serve as a valuable non-invasive tool for predicting patient outcomes and guiding treatment decisions in RCC [105].

Drljevic-Nielsen et al. investigated the potential of spectral dual-layer detector CT parameters as imaging biomarkers in patients with mRCC who were treated with TKIs and immunotherapies. Their first study aimed to assess whether early reduction in spectral dual-layer CT parameters could serve as favorable prognostic indicators. The results demonstrated that patients who showed early reduction in CT parameters, such as iodine concentration and effective atomic number, had significantly longer progression-free survival and overall survival. Drljevic-Nielsen et al.’s second study assessed various CT parameters to determine their association with patient outcomes. This study revealed that certain CT parameters, such as iodine concentration and virtual monochromatic images at specific energy levels, were significantly associated with progression-free survival and overall survival in mRCC patients receiving different therapies. These findings suggest that both pretreatment and spectral dual-layer CT parameters have prognostic utility and could potentially aid in risk stratification and treatment decision making in mRCC patients treated with different therapies [97,98].

##### Retrospective Studies 

In a study by Hall et al. the researchers investigated the prognostic significance of radiological response heterogeneity in mRCC patients treated with VEGF-targeted therapy. The study aimed to determine whether the variation in radiological response within individual patients could predict their overall survival. The findings revealed that higher radiological response heterogeneity was associated with poorer outcomes and shorter survival in mRCC patients. These results highlight the importance of considering the heterogeneity of radiological response as a prognostic factor when evaluating treatment response and predicting patient outcomes in mRCC patients undergoing VEGF-targeted therapy [106].

Go et al. conducted a study aimed at developing a response classifier for VEGFR TKI treatment in mRCC. They analyzed the expression of various biomarkers, including VEGFR and other molecular factors, and developed a response classifier to predict treatment response to VEGFR-TKI therapy. The findings provide insights into the molecular characteristics associated with treatment response, potentially enabling the identification of patients who are more likely to benefit from VEGFR-TKI therapy in mRCC [107].

Mytsyk et al. investigated the usefulness of the apparent diffusion coefficient (ADC) derived from diffusion-weighted magnetic resonance imaging (DW-MRI) in predicting early therapeutic response in mRCC patients. By analyzing the ADC values before and after systemic treatment, they found that changes in ADC values were correlated with treatment response. These results suggest that DW-MRI, specifically the ADC parameter, can serve as a useful tool for predicting early therapeutic response in mRCC patients, facilitating treatment monitoring and decision making [108].

In another study, Wu et al. aimed to assess the response to anti-angiogenic targeted therapy in pulmonary mRCC using the R2* value as a predictive biomarker. They utilized magnetic resonance imaging (MRI) to measure the R2* value, which reflects the concentration of deoxyhemoglobin and serves as an indicator of tumor angiogenesis and hypoxia. The study demonstrated that the baseline R2* value was associated with treatment response and overall survival in mRCC patients receiving anti-angiogenic therapy. These findings suggest that the R2* value could be a predictive biomarker for assessing treatment response in pulmonary metastatic RCC patients undergoing anti-angiogenic targeted therapy [109].

Mittlmeier et al. investigated the utility of 18F-PSMA-1007 PET/CT for response assessment in patients with metastatic renal cell carcinoma undergoing TKI or checkpoint inhibitor therapy. The study evaluated the preliminary results of using PSMA-based positron emission tomography/computed tomography (PET/CT) imaging to assess treatment response. The findings indicated that 18F-PSMA-1007 PET/CT showed promising potential in monitoring response to therapy in patients with metastatic RCC. This suggests that PSMA-based PET/CT imaging may serve as a valuable tool for response assessment in RCC patients receiving TKI or checkpoint inhibitor therapy [101].

#### 3.3.3. Combined Biomarkers

##### Retrospective Studies 

Navani et al. conducted a study to assess the imaging response in patients with mRCC treated with contemporary immuno-oncology combination therapies. The study analyzed radiographic imaging data, including computed tomography (CT) scans, of mRCC patients undergoing combination immunotherapy. They evaluated treatment response based on radiographic assessments such as tumor size reduction, stability, or progression. The results showed that patients who achieved a complete or partial response on imaging had significantly improved progression-free survival compared to those with stable disease or disease progression. This study highlights the importance of radiographic imaging in monitoring treatment response and predicting outcomes in mRCC patients receiving combination immuno-oncology therapies [110].

### 3.4. Physiologic Biomarkers

#### 3.4.1. ICI

##### Prospective Studies

Only one prospective study by De Giorgi et al. studies the role of inflammation and BMI on clinical outcome. They found that a normal BMI combined with inflammation tripled the risk of death, suggesting that these biomarkers are critical prognostic factors for OS in patients with RCC treated with nivolumab. In univariate analysis, markers such as SII, NLR, and PLR were able to predict outcome, and on multivariate analyses, SII ≥ 1375, BMI < 25 kg/m^2^, and age ≥ 70 years independently predicted overall survival. SII changes at 3 months also predicted OS [32].

##### Retrospective Studies 

Other retrospective studies have also investigated the relationship between BMI and treatment outcomes in RCC patients receiving immune-checkpoint inhibition therapy. Labadie et al. conducted a study to evaluate the association between BMI, irAEs, and gene expression signatures with resistance to immune-checkpoint inhibitors and patient outcomes in RCC. They found that higher BMI was significantly associated with poorer response to therapy and reduced overall survival. Additionally, specific gene expression signatures were identified that correlated with resistance to immune-checkpoint inhibition. These findings highlight the potential role of BMI and gene expression profiling as predictive biomarkers for immune-checkpoint inhibition therapy in RCC patients, providing insights into patient stratification and treatment optimization [111].

In a related study, Herrmann et al. explored the prognostic value of BMI and sarcopenia, a condition characterized by muscle loss, in predicting outcomes for RCC patients treated with the immune-checkpoint inhibitor nivolumab. They analyzed the variations in BMI and sarcopenia before and after treatment and assessed their impact on treatment response and overall survival. The study revealed that patients with higher baseline BMI and those who experienced a decrease in BMI during treatment had significantly better outcomes, including improved response rates and prolonged survival. Moreover, the presence of sarcopenia at baseline was associated with worse treatment outcomes. These findings suggest that BMI and sarcopenia can serve as potential prognostic indicators for RCC patients undergoing immune-checkpoint inhibitor therapy, aiding in treatment decision making and patient management [112].

Ueki et al. also conducted a retrospective study to investigate the association between sarcopenia, as determined by the psoas muscle index (PMI), and the response to nivolumab in patients with mRCC. The PMI, which measures the cross-sectional area of the psoas muscle normalized to the patient’s height, is a recognized indicator of muscle mass and overall body composition. The study findings revealed that, patients with a lower PMI, indicating sarcopenia, had a significantly poorer response to nivolumab treatment. These results suggest that sarcopenia, as assessed by the PMI, could serve as a potential biomarker for predicting the response to nivolumab in mRCC patients [113].

#### 3.4.2. VEGF TKIs

##### Retrospective Studies 

Several retrospective studies have also investigated the relationship between body mass and treatment outcomes in renal cell carcinoma (RCC) patients receiving TKI therapy. McKay et al. conducted a study to investigate the effect of weight change during treatment with targeted therapy in patients with mRCC. The researchers assessed the impact of weight loss or gain on treatment outcomes and survival. The study revealed that weight loss during treatment was associated with worse overall survival and progression-free survival in patients with mRCC. On the other hand, weight gain was not significantly associated with survival outcomes. These findings suggest that weight change during targeted therapy may serve as a prognostic factor in mRCC, emphasizing the importance of monitoring and managing weight during treatment [114]. Similarly, Ishihara et al. aimed to evaluate the effect of changes in skeletal muscle mass on oncological outcomes in patients with mRCC receiving first-line sunitinib therapy. The study investigated the association between skeletal muscle mass and treatment response, overall survival, and progression-free survival. The results demonstrated that a decrease in skeletal muscle mass during sunitinib therapy was associated with worse oncological outcomes, including shorter overall survival and progression-free survival. These findings highlight the potential role of skeletal muscle mass as a prognostic indicator in mRCC patients undergoing targeted therapy [115].

Others looked at the predictive and prognostic role of the De Ritis ratio. Janisch et al. conducted a study to assess the predictive value of the De Ritis ratio in mRCC patients treated with TKIs. The De Ritis ratio, calculated as the ratio of aspartate transaminase (AST) to alanine transaminase (ALT), is an indicator of liver function. The researchers found that a higher De Ritis ratio at baseline was associated with poorer survival outcomes in mRCC patients treated with TKIs. This suggests that the De Ritis ratio could serve as a potential predictive biomarker for treatment response and prognosis in mRCC patients receiving TKIs [116]. Kang et al. aimed to evaluate the prognostic impact of the pretreatment aspartate transaminase/alanine transaminase (AST/ALT) ratio in patients treated with first-line systemic TKI therapy for mRCC. The study analyzed the association between the AST/ALT ratio and survival outcomes, including overall survival and progression-free survival. The findings revealed that a higher pretreatment AST/ALT ratio was significantly associated with worse survival outcomes in mRCC patients treated with first-line TKIs. This suggests that the AST/ALT ratio may serve as a simple and readily available prognostic marker in guiding treatment decisions and predicting outcomes in mRCC [117].

Zhang et al. investigated the impact of renal impairment on survival outcomes in patients with mRCC treated with TKIs. The study assessed the association between renal impairment, as determined by estimated glomerular filtration rate (eGFR), and overall survival in mRCC patients. The results demonstrated that patients with impaired renal function had significantly worse overall survival compared to those with normal renal function. This highlights the importance of considering renal impairment in the management and treatment of mRCC patients receiving TKIs [118].

Aktepe et al. aimed to evaluate the impact of the albumin-to-globulin ratio (AGR) on survival outcomes of patients with mRCC. The study assessed the association between AGR, calculated as the ratio of albumin to globulin levels, and overall survival in mRCC patients. The findings revealed that a higher AGR was significantly associated with better overall survival in mRCC patients. This suggests that the AGR could serve as a potential prognostic biomarker for survival outcomes in mRCC, providing additional information for risk stratification and treatment decisions [119].

### 3.5. Miscellaneous Biomarkers

Many studies looked at measures of tumor progression and spread as predictive and prognostic markers in ccRCC. Several other markers are also being studied for predictive and prognostic potential, including race, TKI-induced hypertension, RDW levels, and hemoglobin levels. Details regarding these studies can be found in Appendix A. 

#### 3.5.1. VEGF TKIs

##### Retrospective Studies 

Roussel et al. and Martini et al. aimed to find mechanisms to predict response to therapy. Roussel et al. conducted a study aimed at understanding the molecular mechanisms underlying the glandular tropism observed in metastatic ccRCC. By investigating the gene expression profiles of ccRCC tumors with glandular tropism, they identified specific molecular signatures associated with this phenotype. These findings have important therapeutic implications as they provide insights into the potential targets for developing novel treatment strategies specifically tailored to ccRCC with glandular tropism [120]. Martini et al. developed a novel risk-scoring system for mRCC patients treated with cabozantinib, a tyrosine kinase inhibitor. They integrated various clinical and laboratory parameters to develop a predictive model that can identify patients with different risk profiles and provide prognostic information. This risk-scoring system has the potential to assist clinicians in individualizing treatment approaches and optimizing patient outcomes in mRCC [121].

Several studies examined tumor physiology as a biomarker. Shirotake et al. conducted a single institutional study to evaluate early tumor shrinkage as a predictive factor for mRCC in patients undergoing molecular targeted therapy. They found that patients who achieved early tumor shrinkage demonstrated improved progression-free survival and overall survival. Early tumor shrinkage could serve as an early indicator of treatment response, enabling timely modifications in the treatment regimen to optimize outcomes in mRCC patients [122]. Kammerer-Jacquet et al. investigated hilar fat infiltration as a prognostic factor in metastatic ccRCC patients receiving first-line sunitinib treatment. Their study revealed that the presence of hilar fat infiltration was associated with poorer overall survival and progression-free survival. The identification of hilar fat infiltration as a prognostic factor provides valuable information for risk stratification and treatment decision making in metastatic ccRCC patients [123]. Pieretti et al. investigated the association between tumor diameter response and overall survival in patients with metastatic ccRCC. They found that greater tumor diameter response was significantly associated with improved overall survival. The study highlights the importance of tumor diameter response as a potential predictive factor and emphasizes the need for monitoring tumor size changes during the course of treatment in metastatic ccRCC patients [124]. Shi et al. evaluated the prognostic value of the ratio of maximum-to-minimum diameter of the primary tumor in metastatic ccRCC. They found that a higher ratio was associated with worse overall survival. The ratio of maximum-to-minimum diameter of the primary tumor could serve as a simple and accessible prognostic factor in metastatic ccRCC, aiding in risk stratification and treatment decision making [125]. Yildiz et al. examined prognostic factors for survival in mRCC patients with brain metastases receiving targeted therapy. They identified several factors, including Karnofsky performance status, number of brain metastases, and time to brain metastases, as independent predictors of survival. These findings contribute to the understanding of prognostic factors specific to mRCC patients with brain metastases, aiding in treatment planning and patient management [126].

In summary, these studies provide valuable insights into various aspects of mRCC, ranging from molecular underpinnings and therapeutic implications to prognostic factors and predictive biomarkers. The identification of molecular signatures associated with glandular tropism in ccRCC offers potential targets for personalized treatment strategies. Early tumor shrinkage and tumor diameter response serve as predictive factors for treatment response and overall survival, emphasizing the importance of monitoring tumor size changes during therapy. Prognostic factors such as hilar fat infiltration, the ratio of maximum-to-minimum diameter of the primary tumor, and specific clinical parameters contribute to risk stratification and treatment decision making in mRCC patients. These findings contribute to refining treatment approaches, improving patient outcomes, and advancing our understanding of the complex nature of metastatic renal cell carcinoma.

##### Artificial Intelligence and Multi-Omics Approaches

There has been a recent focus on the ability to apply artificial intelligence and multi-omics approaches to studying a range of cancers and other clinical conditions. Outside of ccRCC, tumor microarrays followed by image analysis and unsupervised learning with machine learning models has shown the ability to predict bladder cancer chemotherapy response [127]. Quantitative imaging followed up with computational analysis of immunohistochemistry revealed characteristic cancer cell profiling in pancreatic cancer [128]. Imaging mass cytometry with follow-up uniform manifold approximation and projection shows the ability to identify tumor biomarkers and predict patient outcomes for ccRCC [129]. Supervised machine learning approaches including convolutional neural networks, logistic regression, and support vector machine also show the ability to predict survival in ccRCC through hematoxylin and eosin histopathological image analysis [130,131]. Combining pathomic and genomic data also demonstrated the ability to stratify risk of ccRCC patients [132]. Due to the nascency of these methods, there is little research done applying these methods in involving prospective clinical trials or in response to specific forms of systemic therapy, but these methods will likely play a large role in shaping personalized and precision medicine.

## 4. Conclusions

The studies reviewed in this article provide important insights into the prognostic factors, treatment response, and response assessment in mRCC patients undergoing targeted therapies. These studies highlight the significance of various clinical, pathological, and radiological factors in predicting patient outcomes and tailoring treatment strategies. Factors such as C-reactive protein levels, lymphocyte microRNA signature, adenosine 2A receptor expression, EVI1 expression, and radiological response heterogeneity have been identified as potential prognostic markers and predictors of treatment response. Additionally, the use of advanced imaging techniques, such as PSMA-based PET/CT, shows promise in assessing treatment response in mRCC patients. The collective findings underscore the importance of a comprehensive approach to patient evaluation, incorporating multiple factors and innovative tools, to optimize the management of mRCC and improve patient outcomes. Further research and validation of these findings are necessary to refine risk stratification, guide treatment decisions, and ultimately enhance the care of patients with metastatic renal cell carcinoma.

While the studies discussed provide valuable insights into prognostic factors, treatment response, and response assessment in mRCC patients, certain limitations should be acknowledged. Firstly, the studies mentioned are based on retrospective analyses or small sample sizes, which may introduce selection bias and limit the generalizability of the findings. Larger prospective studies are needed to validate and further explore the identified prognostic factors and response predictors. Secondly, the studies predominantly focus on specific targeted therapies, such as VEGF-targeted therapy, TKIs, and immune checkpoint inhibitors. The results may not fully represent the entire spectrum of treatment options available for mRCC, including emerging therapies and combination approaches. Therefore, the findings should be interpreted within the context of the specific treatments studied. Additionally, while the studies assess various factors and biomarkers, the complex nature of mRCC and its heterogeneity suggest that multiple factors likely contribute to treatment response and patient outcomes. It is important to consider the interplay between different biomarkers, clinical characteristics, and tumor biology to obtain a comprehensive understanding of prognostic factors in mRCC.

Furthermore, the studies primarily rely on retrospective data or imaging modalities for response assessment, which may have inherent limitations. Variability in radiological assessments, lack of standardized criteria, and potential discrepancies between radiological and clinical response could impact the accuracy and reliability of response evaluation. Lastly, the studies reviewed have varying follow-up periods, and long-term outcomes and survival data may not be fully captured. Longer follow-up durations are necessary to assess the durability of treatment response and evaluate overall survival in mRCC patients. Considering these limitations, further research with larger cohorts, prospective designs, longer follow-up periods, and comprehensive evaluation of multiple factors is warranted to enhance our understanding of mRCC prognosis, treatment response, and response assessment.

## 5. Future Directions

The studies discussed in the previous paragraphs provide valuable insights into prognostic factors, treatment response, and response assessment in mRCC patients. Moving forward, there are several potential directions for future research in this field.

Conducting large-scale prospective studies with standardized protocols and longer follow-up periods can provide more robust evidence on prognostic factors, treatment response, and survival outcomes in mRCC patients. These studies can help validate the findings from retrospective analyses and further explore additional factors that may impact patient outcomes.Continued research is needed to identify and validate novel biomarkers that can predict treatment response, prognosis, and therapeutic resistance in mRCC. The integration of genomic, proteomic, and immunological markers may offer a comprehensive approach to understand the underlying mechanisms and develop personalized treatment strategies.Investigating the efficacy and safety of combination therapies, including targeted agents, immunotherapies, and other emerging treatment modalities, is crucial. Studying the synergistic effects of different therapeutic approaches and identifying predictive markers for optimal treatment selection can improve patient outcomes.Exploring novel response assessment techniques, such as functional imaging modalities (e.g., dynamic contrast-enhanced MRI, diffusion-weighted imaging) and liquid biopsies (e.g., circulating tumor DNA, exosomes), may provide more accurate and timely evaluation of treatment response. Developing standardized criteria and guidelines for response assessment in mRCC can enhance comparability and facilitate clinical decision making.Supplementing traditional clinical trials with real-world evidence from diverse patient populations and clinical settings can provide a more comprehensive understanding of treatment outcomes and enable personalized treatment decisions. Large-scale observational studies and data registries can help assess the effectiveness and safety of therapies in real-world clinical practice.Including patient-reported outcomes, quality-of-life assessments, and patient preferences in research studies can provide a holistic perspective on treatment efficacy and impact on patients’ lives. Understanding the patient experience and incorporating patient-centered endpoints can guide treatment decisions and improve the overall care of mRCC patients.

By focusing on these future directions, researchers can continue to advance our knowledge of mRCC, refine treatment strategies, and ultimately improve patient outcomes in this challenging disease.

## Figures and Tables

**Figure 1 cancers-15-04934-f001:**
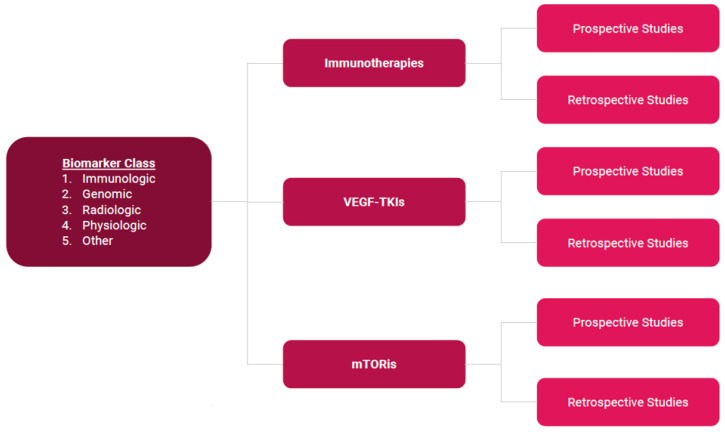
Organization of Review Article.

**Table 1 cancers-15-04934-t001:** Important Definitions.

Term	Definition
Biomarker	A measurable substance whose presence is indicative of disease, infection, or environmental exposure.
Diagnostic Biomarker	Detects or confirms the presence of a disease or condition of interest or identifies an individual with a subtype of the disease.
Predictive Biomarker	Predicts an individual or group of individuals more likely to experience a favorable or unfavorable effect from the exposure to a medical product or environmental agent
Prognostic Biomarker	Identifies the likelihood of a clinical event, disease recurrence, or disease progression in patients with a disease or medical condition of interest
Therapeutic Monitoring Biomarker	Assesses the status of a disease or medical condition for evidence of exposure to a medical product or environmental agent, or to detects an effect of a medical product or biological agent

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
