# Peer review of "Comprehensive Systematic Review of Biomarkers in Metastatic Renal Cell Carcinoma: Predictors, Prognostics, and Therapeutic Monitoring"

_cancers, 2023, doi:10.3390/cancers15204934_

Round 1

Reviewer 1 Report

In this study, the authors conducted comprehensive systematic reviews on the current state of the arts in biomarker discoveries for metastatic renal cell carcinoma. Specifically, the authors categorized biomarkers into the following classes: 1. Immunologic; 2. Genomic; 3. Radiologic; 4. Physiologic; and others. For each class, the authors primarily focused on its applications in immunotherapies, targeted therapies and mTORi. I also commend the authors that further classifications into prospective and retrospective studies were performed, which gives a full context of the agents.

As a computational biologist, I am only able to assess the comprehensiveness of the manuscript from a perspective of biomarker modalities. Recently, there is an increasing interest in using multi-omics data to study anti-tumor efficacy. The advancement of high-dimensional imaging and computational approaches, tumor microenvironment can be dissected at single-cell level and enables biomarker search at an unprecedented resolution. In this study, however, the authors failed to highlight the comprehensiveness and importance of multi-omics. For example, proteomic imaging assay-based pathomics profiling is an integral part of designing novel therapeutics for multiple cancers including renal cell carcinoma [1] [2] [3][4]. Besides, there are also synergistic effects between genomics (as the author suggested) and pathomics in ccRCC prognosis [5]. The role of machine learning/deep learning in integrating multi-omics data has also been highlighted in recent studies [6], therefore a discussion of this aspect specifically for RCC will be a great addition to the story.

Please cite but are not limited to:

[1] Mi, Haoyang, et al. "Quantitative spatial profiling of immune populations in pancreatic ductal adenocarcinoma reveals tumor microenvironment heterogeneity and prognostic biomarkers." Cancer research 82.23 (2022): 4359-4372.

[2] Zhang, Dawei, et al. "Spatial heterogeneity of tumor microenvironment influences the prognosis of clear cell renal cell carcinoma." Journal of Translational Medicine 21.1 (2023): 1-18.

[3] Mi, Haoyang, et al. "Predictive models of response to neoadjuvant chemotherapy in muscle-invasive bladder cancer using nuclear morphology and tissue architecture." Cell Reports Medicine 2.9 (2021).

[4] Cheng, Jun, et al. "Computational analysis of pathological images enables a better diagnosis of TFE3 Xp11. 2 translocation renal cell carcinoma." Nature communications 11.1 (2020): 1778.

[5] Cheng, Jun, et al. "Integrative analysis of histopathological images and genomic data predicts clear cell renal cell carcinoma prognosis." Cancer research 77.21 (2017): e91-e100.

[6] Wessels, Frederik, et al. "Deep learning can predict survival directly from histology in clear cell renal cell carcinoma." Plos one 17.8 (2022): e0272656.

Reviewer 2 Report

This is a well done review of potential prognostic and predictive factors for mRCC, covering large part of the existing data. However, the text is really too long and ripetitive, there is not a sufficient critical approach to the reported data and few, if any, is added to the present literature 

Reviewer 3 Report

The paper is an extensive review of the literature on biomarkers and their role in mRCC. It is well written, clear, very detailed. The work is well organized and comprehensively described.

One minor change: please check in Table 2 second part the title is "Subsequent Clear Cell Histology", while it should be "Subsequent Therapy for Clear Cell Histology".

Round 2

Reviewer 1 Report

The authors have fully addressed my concerns. No further comments.